# Exploring the Attitudes of Pharmacy Students in Saudi Arabia towards Plagiarism Evidence from a Cross-Sectional Study

**DOI:** 10.3390/ijerph192214811

**Published:** 2022-11-10

**Authors:** Salmeen D. Babelghaith, Syed Wajid, Mohamed Al-Arifi, Abdulaziz N. Alotaibi

**Affiliations:** 1Department of Clinical Pharmacy, King Saud University, Riyadh 11451, Saudi Arabia; 2Collage of Pharmacy, King Saud University, Riyadh 11451, Saudi Arabia

**Keywords:** plagiarism, Saudi Arabia, pharmacy, student, attitude

## Abstract

Objectives: The objective of this study was to explore the attitudes towards plagiarism among pharmacy students in Saudi Arabia. Methods: A cross-sectional study was conducted on pharmacy students at King Saud University in Riyadh to assess their attitudes towards plagiarism. The questionnaire consists of 27 items on a 5-point Likert scale (strongly disagree, disagree, neutral, agree, and strongly agree) that assess three attitudes about plagiarism (positive and negative attitudes, as well as subjective norms). Results: A total of 415 pharmacy students took part in this research. Among the whole sample, 55.7% were male, and 44.3% were female. The findings demonstrated a moderately positive attitude towards plagiarism (30 ± 6) and subjective standards (29 ± 7) as well as a moderate negative attitude (22 ± 5). About 26.5% of students did not believe that they worked in a plagiarism-free workplace, and 45.3% of students believed that self-plagiarism should not be punished in the same way as plagiarism. Conclusions: The overall attitudes of pharmacy students were positive. Training in medical writing and research ethics in the undergraduate and graduate pharmacy curricula is mandatory.

## 1. Introduction

Plagiarism comes from the Latin word “plagiarism,” which means “kidnapper” or “plunderer” [1]. It covers the use of published or unpublished data in the original language or as a translation without acknowledging the author. In the field of medicine, scientific and professional activity necessitates a high level of responsibility, absolute devotion, and serious and honest lifetime effort, and unprofessional behavior, such as scientific misconduct, can directly cause errors and harm people’s health and even their lives [1,2]. Plagiarism is not a new problem, nor can it be stated to be exclusive to Saudi universities; rather, it is a major source of concern for the global academic community [1,3]. Plagiarism has received more public attention in the last two decades as a result of the extraordinary expansion of the internet and the concomitant “copy–paste culture” of generation Z students [1]. Plagiarism occurs for a variety of reasons, both intentional and unintentional. These causes can range from carelessness or laziness to a lack of understanding of plagiarism due to a lack of understanding of the issue [4,5]. Online resources are a valuable source of information since they contain a large and diverse amount of information [6,7]. Online resources have become the preferred sources of information since they are easier and take less time to access than journals, books, and encyclopedias. This has resulted in widespread plagiarism, which has become contagious among students [8]. Additionally, language barriers, a lack of moral ethics for intellectual property, and a lack of understanding of what plagiarism entails have all been linked to an increase in the tendency of such behaviors [8]. Although plagiarism is a serious problem, it can be avoided by first identifying the causes of plagiarism and then applying appropriate ways to prevent it, such as training students about plagiarism [4]. Furthermore, we can develop procedures to decrease and eliminate plagiarism in undergraduate students’ research and assignments [4]. Plagiarism is not the same as paraphrasing, and it is one technique to avoid plagiarism. To safeguard copyright, we must reference and cite when we recall someone’s work. Plagiarism is illegal, as previously stated. Although it is a global concern, it is more likely reported in developed countries where research education and training are typically implemented during the undergraduate curricula [4,9]. However, plagiarism is not well-accepted in the majority of underdeveloped countries. For this, there is an urgent need for research on plagiarism in developing nations [8] given that cultural and economic factors may influence how plagiarism is perceived and used. Plagiarism in developing nations may occur for a variety of reasons, including a lack of training, institutional regulations, and oversight on the part of academic institutes and journals; the inadequate development of writing skills and language barriers, however, may be a significant obstacle for many non-native speakers of English [8]. The actual rate of plagiarism among students is unknown, but large-scale studies have claimed that more than 30% of students plagiarize, causing students to be less responsible and self-assured [9].

Plagiarism among Saudi undergraduate students has been the subject of very little published research [1,10]. A study reported that there is a positive tendency toward plagiarism [10]. All these studies were published among medical and dental students. To the best of our knowledge, no studies have been conducted among pharmacy students. Pharmacists are expected to maintain a high level of ethical behavior during their studies, as any unethical action could have a detrimental impact on their future careers. As a result, putting in place policies and processes to teach students about academic dishonesty and how to avoid it is critical [4].

Understanding attitudes toward plagiarism (ATP) is a crucial foundation for both educating and discouraging pupils from plagiarizing [11]. An attitude is a person’s belief, thinking, and emotion towards an item, a group, an event, or a symbol. When individuals have a positive attitude towards something, they will aim to carry it through; however, when they have a negative attitude, they will not [12]. Therefore, if students have a favorable attitude towards plagiarism, they will be more likely to engage in it, while, if students have a negative attitude towards plagiarism, it will force them to write carefully so that they can avoid plagiarizing. It is crucial to have an attitude of disapproval regarding plagiarism. A negative attitude may help students write original work. The goal of this study was to find out how undergraduate pharmacy students at King Saud University in Saudi Arabia felt about plagiarism. We can obtain a baseline picture this way, which will help us improve the education of future practicing pharmacists.

## 2. Materials and Methods

A cross-sectional study was carried out during the three months from 1 September to 30 November 2021. This study used a validated questionnaire. The targeted sample included pharmacy students from King Saud University in Riyadh, Saudi Arabia.

### 2.1. Questionnaire

The students’ attitudes were assessed using the Attitude Toward Plagiarism (ATP) questionnaire. The ATP questionnaire was adopted from a previous study [13]. It is divided into two sections: the first section involves demographic data (gender, education level, and education degree). The second section involves the attitude towards plagiarism, on a scale that assesses three attitudes: positive attitude, negative attitude, and subjective norms. On a 5-point Likert scale, 29 statements were provided, with (1) indicating strongly disagree, (2) indicating disagree, (3) indicating neither agree nor disagree, (4) indicating agree, and (5) indicating strongly agree for each statement. Each response was given a point value, and scores were computed by adding them together. For each factor, the lowest and highest potential scores were determined, and the ranges were divided into three equal sections to represent low, moderate, and high score scales. Plagiarism is accepted and approved with a positive attitude. It was assessed by 12 items with a score range of 12 to 60. A low score range (12–28) reflects a negative attitude towards plagiarism. Positive attitudes regarding plagiarism are mostly measured by statements about procedures that participants perform on their own. Plagiarism is depreciated and condemned when students have a negative opinion of it. It is based on seven statements, each with a score ranging from 7 to 35. In this attitudinal aspect, a high score range of 27–35 suggests a no-tolerance attitude toward plagiarism. Negative attitude statements are primarily related to procedures carried out by others or to society as a whole. In two steps, the questionnaire was validated. First, a research expert in the associated field reviewed the first draft. Second, a pilot study was conducted with a randomly selected sample of 25 participants to obtain their feedback on how to make the questionnaire more user-friendly. The Cronbach’s alpha was estimated for reliability tests, and a score of 0.8 suggested that the questionnaires were useful for the study.

### 2.2. Data Analysis

The Statistical Package for the Social Sciences (SPSS) (version 26 for Windows (SPSS Inc., Chicago, IL, USA) was used to analyze the data. The demographic characteristics were summarized using descriptive statistics. To examine the difference in the variables, univariate analysis (chi-squared test/Fisher’s exact test) was utilized. All statistical tests were conducted using a 0.05 significance level.

## 3. Results

A total of 415 pharmacy students took part in this research. Among the whole sample, 55.7% were male, and 44.3% were female. Respondents from Pharm D represented the majority of the sample (68.7%). The majority of responses (29.4%) were from those in the fourth year, followed by the sixth (20.2%), fifth (17.1%), third (15.7%), second (11.6%), and first years of education (6.0%) as showed in Table 1. An analysis of numerous levels of the ATP questionnaire was carried out (Table 2). This study found that students had a moderate attitude towards plagiarism. The average score for those who had a positive attitude towards plagiarism was 30 ± 6, showing that a significant percentage of students rationalize and condone plagiarism. Additionally, students showed a moderate level of a negative attitude towards plagiarism (22 ± 5). In addition, subjective norms were modest, showing that students do not view plagiarism in the community in a positive light.

Table 3 shows the distributions of responses by the questionnaire statement, and several of the facts in the table warrant special attention.

The findings demonstrated that about 34% of the students would plagiarize to hide their lack of writing abilities (statement 1). More than one-third of the students considered self-plagiarism to be innocuous and unpunishable (items 3, 4, and 6). When it came to the scientific method, academic time, and plagiarism, the majority of students had no clear attitude (neutral). About 34 % of students believed they had the right to plagiarize to some extent because they were still in the process of learning (statement 7).

Unfortunately, nearly 40% of students believed that plagiarism impoverishes the inquiring spirit (statement 11), and almost 56% of students said that plagiarism is an essential topic to debate (statement 12). A total of 22% of students believe that plagiarism is not a serious offense like the appropriation of material assets (statement 13) and that it has no negative impact on science (statement 15). A majority of students (73.1 %) considered the punishment for plagiarism to be the same as stealing an exam (statement 17), as presented in Table 4.

Plagiarism was endorsed by more than half of the respondents. A few students (26%) stated that they work in a plagiarism-free environment (statement 24). About 35% of students considered it ethically bad to copy text from their earlier work (statement 21). Students were split on whether plagiarizing is necessary to gain time for other activities (statements 25 and 26)—they agreed, disagreed, and were neutral in equal percentages. (Table 5)

## 4. Discussion

To the best of our knowledge, this study is the first to document pharmacy students’ attitudes toward plagiarism in Saudi universities. The goal of this study was to determine the level of attitude of pharmacy students regarding plagiarism. The slight disparity in the number of males (231 students) and girls (184 students) that participated mirrored the reality that males make up the majority of students enrolling in pharmacy schools in Saudi Arabia. A case study was conducted among Saudi undergraduate and postgraduate students that aimed to assess their attitudes towards cheating and plagiarism and found that the respondents had a high level of awareness of plagiarism [14], similar to the previous study [15]. On the other hand, a study conducted on Saudi Arabian physicians in training discovered that high percentages of the participants had an aversion to plagiarism as well as a tendency towards a reduced tolerance of plagiarism and a preference for the personal approval of plagiarism practice in society [10]. In Croatia, a study involving 144 students from the College of Pharmacy and Medical Biochemistry was conducted to analyze the ATP, and the results revealed that 90% of the participants had a generally negative ATP by agreeing that such behavior would negatively damage the investigator’s spirit. Nevertheless, 60% of students said that plagiarism was not a severe problem and that it would have no impact on science [13]. There is a lack of competence in scientific methodology and writing skills, followed by inadequate knowledge of academic integrity. Students believe that it is often impossible to avoid using other people’s words without citing the source, translation from a foreign language is appropriate, and using the same methodology from earlier studies is justified. Inadequate knowledge of scientific techniques can be detected in students’ attitudes; they believe that it is acceptable to plagiarize if the article is of high scientific worth. Our findings are in line with those of other studies. These findings are similar to those of a study conducted in Croatia that found that the majority of pharmacy students felt the same way [13]. However, many studies carried out among medical students reported that, in most cases, plagiarism is due to a lack of instruction in research methods and reference practices [16,17,18].

Several studies have revealed that a major reason for plagiarizing is the lack of punishment for plagiarized work because students believe plagiarized work should not be considered a serious academic offense. Three studies suggested that Saudi Arabian higher education be subjected to stronger punishments and rules as a result of the data from all three sources at the university [1,3,19]. Despite this, several countries have strong professional sanctions for plagiarism, ranging from verbal warnings to the resubmission of work, credential withdrawal, failing semesters, suspension, and expulsion [2]. In the current study, the mean scores for the positive domain, negative domain, and subjective norms of the ATP scale were 30 ± 6, 22 ± 5, and 29 ± 7, respectively. A study also found that moderate positive attitudes and subjective norms about plagiarism were associated with moderate to high negative attitudes (36 ± 7, 26 ± 4, and 32 ± 6, respectively) [13]. Another two studies conducted in Saudi Arabia among medical students who used the same ATP reported that the students had moderate scores in all three domains [10,16]. About 50% of students say that plagiarism impoverishes the inquiring spirit. About 22% of students, on the other hand, agree that plagiarized papers do not hurt science. It is alarming that 23.1% of students agree and 34% are undecided about whether plagiarizing is bad or not. Our findings are in line with those of other studies. It was a study that looked at students’ awareness and knowledge of plagiarism.

In addition, this study found that most students believed that self-plagiarism should not be punished in the same manner as plagiarism, 73.9% of students said that plagiarism is the equivalent of stealing an exam, and 22.4% of students, on the other hand, agree that plagiarized papers do not hurt science. Additionally, most of the students (36.3%) thought that young researchers should receive a lesser punishment for plagiarism. To decrease the prevalence of plagiarism, we must first understand its origin. The majority of medical students and their faculties admitted to plagiarizing at least once throughout their lives in recent research [20]. This confirms our results: only 26.5% of pharmacy students agreed that they worked in a plagiarism-free environment. The reasons for this heinous practice in Saudi health schools are several. Stakeholders should develop appropriate policies, and research and medical writing training modules should be included in pharmacy curricula. The Higher Education Commission of Saudi and the Saudi Commission for Health Specialties should assist medical colleges and universities in training and establishing ethics review committees and institutional review boards. Students, trainees, and faculty at medical schools should have access to plagiarism detection software. Those attending medical research ethics or medical writing courses, as well as those with past publications, have all investigated plagiarism in various studies. Medical students who received medical writing instruction exhibited an overall negative attitude towards plagiarism, whereas those who received research training did not. Furthermore, even though a majority of faculty members had previously undergone training in both medical writing and research ethics, there was an overall positive ATP among them [10]. Furthermore, studies on healthcare students in many elements of healthcare behaviors and clinical knowledge-related assessments have been reported [21,22,23,24,25]. However, studies on plagiarism assessment among students were lacking or limited; therefore, this study would add beneficial knowledge in the research area and would serve as a future reference for upcoming studies.

### Limitations

Since this study was conducted in the Saudi Arabian province of Riyadh, extrapolating its findings to other regions may not give an accurate picture of the situation. Additionally, it is advised to conduct a comprehensive survey to allow pharmacists from all regions of Saudi Arabia with a variety of experiences to participate.

## 5. Conclusions

This was the first study that aimed to evaluate the attitude of pharmacy students towards plagiarism in Saudi Arabia. Most pharmacy students in Saudi Arabia were found to have a negative attitude towards plagiarism. These results highlight the importance of good teaching at the beginning regarding the definition, practice, and effects of plagiarism, as well as how it might harm the literature and how to avoid it.

## Figures and Tables

**Table 1 ijerph-19-14811-t001:** Demographic data of participants.

Variables	N (%)
Gender	
Male	231 (55.7)
Female	184 (44.3)
Age groups (years)	
<18	6 (1.4)
18–25	380 (91.6)
26–30	28 (6.7)
31–35	1 (0.2)
Level of education	
First year	25 (6.0)
Second year	48 (11.6)
Third year	65 (15.7)
Fourth year	122 (29.4)
Fifth year	71 (17.1)
Sixth year	84 (20.2)
Degree of Education	
B pharm	111 (26.7)
Pharm D	285 (68.7)
Postgraduate	19 (4.6)

**Table 2 ijerph-19-14811-t002:** Attitude towards plagiarism score and its interpretations of levels.

Attitudinal Determinants	Mean ± SD	Score Range	Level
Positive attitude	30 ± 6	10–23	Low *
		24–37	Moderate
		38–50	High
Negative attitude	22 ± 5	7–16	Low
		17–26	Moderate
		27–35	High *
Subjective norms	29 ± 7	10–23	Low *
		24–37	Moderate
		38–50	High

***** favorable attitude in terms of academic integrity.

**Table 3 ijerph-19-14811-t003:** Distributions of responses to the positive attitude plagiarism questionnaire (N = 415).

Items Expressing a Positive Attitude	1 N (%)	2 N (%)	3 N (%)	4 N (%)	5 N (%)
Sometimes one cannot avoid using other people’s words without citing the source, because there are only so many ways to describe something.	37 (8.9)	70 (16.9)	153 (36.9)	120 (28.9)	35 (8.4)
2. When I do not know what to write, I translate a part of a paper from a foreign language.	64 (15.4)	83 (20)	132 (31.8)	97 (23.4)	39 (9.4)
3. Self-plagiarism is not punishable because it is not harmful (one cannot steal from oneself).	56 (13.5)	84 (20.2)	121 (29.2)	114 (27.5)	40 (9.6)
4. Self-plagiarism should not be punishable as plagiarism.	36 (8.7)	65 (15.7)	125 (30.1)	137 (33)	52 (12.5)
5. It is justified to use one’s own previously published work without providing citations to complete the current work.	87 (21)	120 (28.9)	104 (25.1)	83 (20)	21 (5.1)
6. Young researchers who are just learning the ropes should receive milder punishments for plagiarism.	44 (10.6)	79 (19)	141 (34)	111 (26.7)	40 (9.6)
7. It is justified to use previous descriptions of a method because the method itself remains the same.	35 (8.4)	60 (14.5)	140 (33.7)	145 (34.9)	35 (8.4)
8. If one cannot write well in a foreign language (e.g., English), it is justified to copy parts of a similar paper already published in that language.	77 (18.6)	119 (28.7)	104 (25.1)	83 (20)	323 (7.7)
9. If a colleague of mine allows me to copy from her/his paper, I am NOT doing anything bad, because I have his/her permission.	51 (12.3)	99 (23.9)	113 (27.2)	102 (24.6)	50 (12)
10. Plagiarized parts of a paper may be ignored if the paper is of great scientific value.	55 (13.3)	112 (27)	126 (30.4)	96 (23.1)	26 (6.3)

1 = Strongly disagree, 2 = Disagree, 3 = Neutral, 4 = Agree, and 5 = Strongly agree. Statements expressing a negative attitude

**Table 4 ijerph-19-14811-t004:** Distributions of responses to the negative attitude toward plagiarism questionnaire (N = 415).

Items Expressing a Negative Attitude	1	2	3	4	5
Plagiarism impoverishes the investigative spirit.	35 (8.4)	63 (15.2)	153 (36.9)	121 (29.2)	43 (10.4)
2. In times of moral and ethical decline, it is important to discuss issues such as plagiarism and self-plagiarism.	6 (25)	43 (10.4)	112 (27)	161 (38.8)	74 (17.8)
3. Since plagiarism is taking other people’s words rather than tangible assets, it should NOT be considered very important.	90 (21.7)	127 (30.6)	106 (25.5)	71 (17.1)	21 (5.1)
4. The names of the authors who plagiarize should be disclosed to the scientific community.	38 (9.2)	80 (19.3)	141 (34)	115 (27.7)	41 (9.9)
5. A plagiarized paper does not harm science.	97 (23.4)	118 (28.4)	107 (25.8)	74 (17.8)	19 (4.6)
6. Plagiarists do not belong in the scientific community.	39 (9.4)	60 (14.5)	147 (35.4)	116 (28)	53 (12.8)
7. Plagiarizing is as bad as stealing an exam.	41 (9.9)	67 (16.1)	0	128 (30.8)	179 (43.1)

1 = Strongly disagree, 2 = Disagree, 3 = Neutral, 4 = Agree, and 5 = Strongly agree. Items expressing subjective norms.

**Table 5 ijerph-19-14811-t005:** Distributions of responses to the subjective norms attitude toward plagiarism questionnaire (N = 415).

Items Expressing Subjective Norms	1	2	3	4	5
Those who say they have never plagiarized are lying.	0	130 (31.3)	143 (34.5)	101 (24.3)	41 (9.9)
2. Sometimes I copy a sentence or two just to become inspired for further writing.	28 (6.7)	59 (14.2)	118 (28.4)	152 (36.6)	58 (14)
3. Authors say they do NOT plagiarize, when in fact they do	34 (8.2)	149 (35.9)	0	193 (46.5)	39 (9.4)
4. I do NOT have a bad conscience for copying verbatim a sentence or two from my previous papers.	41 (9.9)	87 (21)	141 (34)	114 (27.5)	32 (7.7)
5. Sometimes I am tempted to plagiarize because everyone else is doing it (students, researchers, physicians).	68 (16.4)	92 (22.2)	116 (28)	111 (26.7)	28 (6.7)
6. It is NOT so bad to plagiarize.	93 (22.4)	118 (28.40	108 (26)	72 (17.3)	24 (5.8)
7. I work (study) in a plagiarism-free environment.	57 (13.7)	97 (23.4)	151 (36.4)	86 (20.7)	24 (5.8)
8. Sometimes, it is necessary to plagiarize.	46 (11.1)	101 (24.3)	124 (29.9)	102 (24.6)	42 (10.1)
9. Plagiarism is justified if I currently have more important obligations or tasks to complete.	60 (14.5)	101 (24.3)	140 (33.7)	90 (21.7)	24 (5.8)
10. I keep plagiarizing because I have not been caught yet.	119 (28.7)	1023 (24.60	117 (28.2)	57 (13.7)	20 (4.8)

1 = Strongly disagree, 2 = Disagree, 3 = Neutral, 4 = Agree, and 5 = Strongly agree.

## Data Availability

The datasets used and analyzed during the current study are available from the corresponding author on reasonable request.

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
