# Peer review of "Exploring the Attitudes of Pharmacy Students in Saudi Arabia towards Plagiarism Evidence from a Cross-Sectional Study"

_ijerph, 2022, doi:10.3390/ijerph192214811_

Round 1
Reviewer 1 Report
Dear authors,
Thank you for an interesting read. I agree that plagiarism is a subject area that needs to be investigated and considerations made on how to improve students’ approach to it. However, and generally speaking, I think the focus of this paper should be redirected.
The abstract itself needs to be reworked, especially with a conclusionary sentence such as “insufficient level of concern” does not emphasise the importance of this work.
Throughout the text, there are grammatical errors e.g. punctuation, poor phrasing, long sentences
Abbreviated term i.e. Plg were never defined within the text.
The introduction discusses the importance of plagiarism. However, you have not distinguished between positive, negative and subjective plagiarism. Further, it is widely known that plagiarism is a more common offence in under-developed countries. However, you have not provided any attempt to produce a literature review to explain why. Therefore, more investigation is required to strengthen the introduction and to emphasise the relevance and importance of this work. You may also wish to refer to the paper by Sales 2020 which discusses that although students in an underdeveloped country are aware of plagiarism, they chose to ignore it regardless. Further, it should be highlighted that this is a case study
It was nice to see how the study was piloted and validated. However, there are a number of texts throughout the article that are not supported with references.
The percentage of 1st year students needs to be included in line 111
The discussion should encompass a greater discussion on why 1/3 of students think plagiarism is innocuous and unpunishable. Further no limitations of this study have been included e.g. the small sample size etc.
Thank you
Author Response
you for an interesting read.
I agree that plagiarism is a subject area that needs to be investigated and considerations made on how to improve students’ approach to it. However, and generally speaking, I think the focus of this paper should be redirected.
The abstract itself needs to be reworked, especially with a conclusionary sentence such as “insufficient level of concern” does not emphasize the importance of this work.
Answer: We appreciate your comment. We rewrite the conclusion. The overall attitudes of pharmacy students were positive. It is mandatory for training in medical writing and research ethics in the undergraduate and graduate pharmacy curricula.
Throughout the text, there are grammatical errors e.g. punctuation, poor phrasing, long sentences.
Answer: The entire manuscript was revised by native speaker
Abbreviated term i.e. Plg were never defined within the text.
Answer: We appreciate your careful note thank you. it is removed and written the full term
The introduction discusses the importance of plagiarism. However, you have not distinguished between positive, negative and subjective plagiarism. Further, it is widely known that plagiarism is a more common offence in under-developed countries. However, you have not provided any attempt to produce a literature review to explain why. Therefore, more investigation is required to strengthen the introduction and to emphasize the relevance and importance of this work. You may also wish to refer to the paper by Sales 2020 which discusses that although students in an underdeveloped country are aware of plagiarism, they chose to ignore it regardless. Further, it should be highlighted that this is a case study.
Answer: Yes, we agree with your comment. Most reports about plagiarism come from developed countries while in developing counties no much reports were recommended. There is an urgent need for research on plagiarism in developing nations (11), given that cultural and economic factors may influence how plagiarism is perceived and used. Plagiarism in developing nations may occur for a variety of reasons, including a lack of edu-cation about research ethics, inadequate writing skills development, tolerance for inappropriate behavior in academic and professional settings as well as absence of institutional policies.
Understanding attitudes toward plagiarism (ATP) is a crucial foundation for both educating and discouraging pupils from plagiarizing (Kumari) (13). attitude is a person’s belief, thinking, and emotion toward item, a group, an event, or a symbol. When individuals have a positive attitude toward something, they will aim to carry it through, however when they have a negative attitude, they will not (Rodhiya et al.,2020). Therefore, if students have a favorable attitude about plagiarism, they will be more likely to engage in it. While, if students have a negative attitude about plagiarism, it will force them to write carefully so that they can avoid plagiarizing. It's crucial to have an attitude of disapproval regarding plagiarism. A negative attitude may help the students to write an original work.
It was nice to see how the study was piloted and validated. However, there are a number of texts throughout the article that are not supported with references.
Answer: again thank you very much for your carful note. the reference was added (please see method section Ref. No 13)
The percentage of 1st year students’ needs to be included in line 111
Answer: we appreciate your comment: the percentage was included
The discussion should encompass a greater discussion on why 1/3 of students think plagiarism is innocuous and unpunishable.
Answer: This study found that about 37.1% of pharmacy students believed that self-plagiarism should not be punished. This could be due to most of students don’t know what plagiarism is, the need to achieve "excellence" in the work and academic dishonesty
Further no limitations of this study have been included e.g. the small sample size etc.
Limitations
Since this study was conducted in the Saudi Arabian province of Riyadh, extrapolating its findings to other regions may not give an accurate picture of the situation.
Additionally, it is advised to conduct a comprehensive survey to allow pharmacists from all regions of Saudi Arabia with a variety of experiences to participate.
Reviewer 2 Report
This is an interesting study, and relevant to pharmacy education as well as the literature on plagiarism.
I have a question I think should be explored. 73.9% of students agreed that plagiarism is equivalent to stealing an exam. However, the conclusion was that the students showed an insufficient level of concern about plagiarism. So what does this indicate about students' thoughts about stealing an exam. The question did not indicate that stealing an exam was bad or unethical, so are students not concerned about stealing exams as they considered plagiarism and stealing exams to be equivalent? This should be explored further.
Also, descriptive statistics were used. Inferential statistics would strengthen the results and the validity of inferences for readers.
Author Response
this is an interesting study, and relevant to pharmacy education as well as the literature on plagiarism.
I have a question I think should be explored. 73.9% of students agreed that plagiarism is equivalent to stealing an exam. However, the conclusion was that the students showed an insufficient level of concern about plagiarism. So what does this indicate about students' thoughts about stealing an exam. The question did not indicate that stealing an exam was bad or unethical, so are students not concerned about stealing exams as they considered plagiarism and stealing exams to be equivalent? This should be explored further.
Answer: We appreciate your comment. When asked students about their thoughts of is plagiarizing consider bad as stealing an exam? most of them agreed with that.
Academic dishonesty includes both plagiarism and cheating, in this study 73.9% of students said that plagiarism is the equivalent of stealing an exam. This means that most of students believed that plagiarism is most serious and unethical as well as cheating behaviors (stealing an exam).
Also, descriptive statistics were used. Inferential statistics would strengthen the results and the validity of inferences for readers.
Answer: We appreciate your comment. However, this study aimed to assess the attitude of pharmacy students. For this we used descriptive and some variation analysis such as chi square, inferential statistics could be applied in future studies to explore the factors influence their attitude.
Thank you very much.